# Controlled Silanization of Transparent Conductive Oxides as a Precursor of Molecular Recognition Systems

**DOI:** 10.3390/ma16010309

**Published:** 2022-12-29

**Authors:** Anna Domaros, Dorota Zarzeczańska, Tadeusz Ossowski, Anna Wcisło

**Affiliations:** Department of Analytical Chemistry, Faculty of Chemistry, University of Gdansk, ul. Wita Stwosza 63, 80-308 Gdansk, Poland

**Keywords:** conductive materials, FTO electrodes, electrochemical measurements, silanization, electrode modification

## Abstract

The search for new molecular recognition systems has become the goal of modern electrochemistry. Creating a matrix in which properties can be controlled to obtain a desired analytical signal is an essential part of creating such tools. The aim of this work was to modify the surface of electrodes based on transparent conductive oxides with the use of selected alkoxysilanes (3-aminopropyltrimethoxysilane, trimethoxy(propyl)silane, and trimethoxy(octyl)silane). Electrochemical impedance spectroscopy and cyclic voltammetry techniques, as well as contact angle measurements, were used to determine the properties of the obtained layers. Here, we prove that not only was the structure of alkoxysilanes taken into account but also the conditions of the modification process—reaction conditions (time and temperature), double alkoxysilane modification, and mono- and binary component modification. Our results enabled the identification of the parameters that are important to ensure the effectiveness of the modification process. Moreover, we confirmed that the selection of the correct alkoxysilane allows the surface properties of the electrode material to be controlled and, consequently, the charge transfer process at the electrode/solution interface, hence enabling the creation of selective molecular recognition systems.

## 1. Introduction

In recent years, materials chemistry has become one of the most rapidly developing fields of science. It mainly comprises the synthesis of new electrode materials or the modification of their surfaces to obtain specific properties and increase their applicability [1,2,3,4,5,6,7,8].

Currently, there is growing interest in scientific research focusing on semiconductor electrode materials based on oxide and carbon layers, which are characterized by various electrical properties [9,10,11,12,13]. However, the combination of two different properties characteristic of transparent conductive oxides, i.e., optical transparency and high electrical conductivity, makes this material extremely interesting for further research [14,15]. Fluorine-doped tin oxide (FTO) electrodes are considered a very promising material due to their greater stability in atmospheric conditions and higher temperature resistance than indium tin oxide (ITO) electrodes. Moreover, this material is chemically inert, mechanically resistant, and has high resistance to physical abrasion [16,17]. Both FTO and ITO electrodes find practical applications in a wide range of devices, among others, to provide transparent conductive coatings for use in touch panels, flat screens, aircraft cockpit windows, or plasma monitors. Thin oxide films are also used in the production of organic light-emitting diodes (OLEDs) and in solar cells [18,19]. In addition, due to their unique features, these materials provide a good basis for the modification of surface properties [16,20,21,22,23,24]. Moreover, they are also used in classical electroanalytical measurements as working electrodes, in the determination of a wide range of electroactive compounds [25].

A specific molecular recognition system can be synthesized by depositing systems on their surface with specific reducing and oxidizing properties (e.g., quinone derivatives), compounds showing the ability to form hydrogen bonds (e.g., edetic acid derivatives), coordination and donor systems (e.g., bipyridyl, tetraxetane), or biomolecules (e.g., proteins, enzymes). Thanks to these properties, FTO electrodes are used in biosensors, as well as in classical electroanalytical measurements concerning the determination of redox-active compounds [26,27,28,29]. Organosilicon monolayers bind most effectively to a surface with a silicon atom in its structure [30,31,32]. In the literature, one can find plenty of reports on the modification of ITO- and FTO-type glass electrodes with silane compounds. Thus, the electrode material also shows excellent silane-binding efficiency, due to the presence of surface inorganic oxides deposited in the easily oxidized form of a thin layer on the glass [20,21,23,33].

The covalent attachment of specific biomolecules to the surface of a conducting material is of great importance for the development of sensors based on molecular recognition. The functionalization of a surface by inclusion in the structure of a compound with specific properties requires the presence of an intermediate organic layer. In the case of oxide surfaces, such as an FTO electrode, organosilanes are the most common compounds and mediate in the anchoring of molecules to the substrate [9,34,35,36,37,38]. The silanization process is the first step toward the construction of a biosensor based on the FTO electrode. Reports available in the literature show a wide variety of conditions used at this stage, starting from the choice of the solvent and silane concentration to the temperature and time of electrode incubation in the silane solution [23,24,30,39]. However, the articles do not provide details as to how these specific conditions affect the properties of the obtained layers, and, thus, what parameters to choose to achieve the desired effect. Therefore, we decided to optimize the deposition conditions of the selected group of alkoxysilanes and their mixtures, paying particular attention to the settling time and the reaction temperature [40]. Focusing on the selection of appropriate conditions, our aim was to obtain the best possible repeatability of the silane deposition process for the FTO electrode surface. In addition, we verified the stability of the modified electrodes to determine their working time and, consequently, their susceptibility to further modification stages.

The aim of the presented work was the synthesis of organic films on conductive electrode materials and the characterization of their properties, which are important for electrochemical applications in the context of creating new molecular recognition systems.

Bearing in mind the set goal, we decided to compare the properties of the currently manufactured electrodes, with a particular emphasis on the electrochemical parameters and their modification potential. Based on previous research, we were able to select such a material—FTO—which met the expected requirements [40]. Subsequently, we modified the materials with alkoxysilanes to provide them with new properties. Modifying the electrode surfaces with chemicals can be a simple but efficient way to obtain a material with the specific properties desired for a given application. Thanks to the appropriate selection of modifiers and the skillful handling of the parameters of their combination, the scope of application of a given electrode is significantly extended. The possibilities are endless as one does not have to set the limits to just one modification stage. More and more often, extensive molecular recognition systems are being prepared in which specific functional groups that can act as an electrochemical probe are attached to linker compounds. The issue that should be taken into account, however, is the appropriate and reliable characterization of the obtained surface, so that it can serve as an innovative solution when researching problems in the future.

## 2. Materials and Methods

Preparation of the electrode surface: Cleaning the surface of the FTO electrodes began with sonication of the plates in three successive solvents: in acetone and ethyl alcohol for 5 min and in distilled water for 10 min. The electrodes were then dried at 70 °C for 60 min and immersed in an “alkaline piranha” solution for about two hours. The temperature was maintained at 60 °C during the process. The aim of this treatment was to activate the electrode surface by oxidizing the groups on its surface to silanol groups. Subsequently, the electrodes were rinsed with distilled water and placed back in the oven at 70 °C for about 30 min [41,42].

### 2.1. Modification Procedure

Silanization with alkoxysilanes: FTO electrodes with cleaned and activated surfaces were immersed in 2.5% (m/m) alkoxysilane solutions, which were prepared immediately before this step in 96% ethyl alcohol. Table 1, below, shows the names and formulas of the silanes used for modification.

The silanizing solutions consisted of one or a mixture of two silanes. The modifications were carried out under various process conditions. The times of electrode incubation in solutions and the temperature at which the silanization reaction was carried out varied. The experimental conditions are shown in Table 2. After the specified time had elapsed (16 or 72 h), the electrodes were rinsed several times with ethanol to remove the excess silanes. The modified electrodes were then heated in an oven at 120 °C for 2 h.

The modified FTO electrodes listed in Table 2 were further silanized (in a second reaction) according to the same procedure, to yield four new electrodes.

Three alkoxysilanes were used to functionalize the FTO substrates: 3-aminopropyltrimethoxysilane (APTMS), trimethoxypropylsilane (PTMS), and trimethoxyoctylsilane (OTMS), differing in the structure of the side chain (Table 1, Figure 1). The electrochemical properties of the modified FTO/APTMS, FTO/OTMS, and FTO/APTMS_OTMS electrodes were compared to those of the unmodified FTO electrode. Additionally, we assessed how the presence of the amino group in the side chain of the alkoxysilane (FTO/APTMS) changed the surface properties of the electrode, compared to its alkyl derivative (FTO/PTMS). We verified whether the long alkyl chain in the trimethoxysilane structure (FTO/OTMS) hinders the transfer of electrons at the electrode/solution interface and whether the deposition of a mixture of silanes (FTO/APTMS_OTMS) on the surface of the FTO electrode would show better conductive properties than their one-component counterparts (FTO/APTMS and FTO/OTMS). In addition, we examined whether the order of silane deposition is important in the case of a two-step silanization of the FTO electrode surface (e.g., FTO/APTMS/OTMS and FTO/OTMS/APTMS). We also assessed the changes in the surface wettability on the FTO/APTMS, FTO/PTMS, and FTO/OTMS electrodes.

### 2.2. Electrochemical Measurements

For the electrochemical evaluation of the modified FTO electrode surfaces, we employed impedance spectroscopy and cyclic voltammetry, which are suitable for the study and monitoring of the changes taking place on the electrode surfaces as a result of the performed silanization reactions. Using the electrochemical impedance spectroscopy (EIS) technique enabled the determination of the charge transfer resistance (Rct), which is a measure of the resistance of self-organizing silane layers to electron transfer, from the solution to the electrode surface. Recorded impedance spectra are displayed in the form of Nyquist diagrams.

The electrochemical measurements were carried out using an apparatus consisting of a Multi Autolab/M204 potentiostat, equipped with an FRA module (Metrohm, Barendrecht, The Netherlands) and a glass measuring cell. The measurements were performed in a Faraday cage, in order to isolate the system from the influence of external factors in a three-electrode measurement system, consisting of a working electrode (unmodified and modified FTO, dimensions 1.5 × 3.5 cm) (Sigma Aldrich, Schnelldorf, Germany), silver chloride reference electrode (Ag/AgCl, 0.1 M KCl) (Mineral, Warsaw, Poland), and platinum wire counter-electrodes (Mennica-Metale Sp. z.o.o., Radzymin, Poland).

Nova 2.11 software was used to operate the potentiostat, as well as to analyze and simulate the electrochemical impedance spectroscopy (EIS) spectra (Metrohm AG, Herisau, Switzerland). OriginPro 2020 software (OriginLab, Northampton, MA, USA) was used for data analysis.

Cyclic voltammetry measurements were performed for a scanning speed of V = 100 mV/s. The range of measurements (initial and final measurements, as well as peak potentials) was adjusted for the best visualization of the processes taking place on the electrode surface. Measurements of EIS were performed in the frequency range from 10 kHz to 100 mHz, with a wave amplitude of 0.01 V and a sinusoidal excitation signal.

Two types of basic electrolytes were used in the electrochemical measurements—0.5 M potassium chloride and 0.5 M sodium sulfate solutions. The model redox system used in the research was potassium hexacyanoferrate (III/II). All solutions were prepared from weights by dissolving them in distilled water in such amounts as to ultimately obtain solutions with a concentration of 0.01 M. The prepared solutions were stored in a refrigerator; however, before the tests, they were brought back to ambient temperature [42].

### 2.3. Contact Angle Measurements

The wettability of the surface of the electrode materials was determined by measuring the contact angle at room temperature, using standard liquids based on the sitting drop static method. The reported contact angle values are average values, measured at different positions on the electrode surface. Drop shape analysis (with a volume of 2 µL or 4 µL) was performed using the circle method and the Young–Laplace method [1,5,40,42,43,44,45]. Measurements were made using the KRÜSS Drop Shape Analyzer—DSA100 apparatus.

## 3. Results and Discussion

### 3.1. Alkoxysilane Modification

The impedance spectrum for the unmodified FTO electrode is characterized by a semicircle in the high-frequency range, while in the low-frequency range, it is characterized by a straight line inclined at an angle of 45° (Figure 2A) [46]. This is a classic example of a spectrum, showing that the electron transfer process for the redox [Fe(CN)_6_]^3−/4−^ pair is quasi-reversible and diffusion-controlled [47]. The measurement of the charge transfer resistance, Rct, is the diameter of the semicircle obtained in the Nyquist plots. The larger the diameter, the greater the charge transfer resistance. As the value of Rct changes, the capacity of the double layer (Q) also changes. Low Q values indicate an increase in material thickness, due to the surface modification. The n parameter, which defines the degree of heterogeneity in the newly formed structure, is inherent in the capacity of the double layer. In the case of an electrode with a perfectly smooth surface, this equals 1. The Rct value, calculated on the basis of the equivalent circuit used, R(Q(RW)), for the FTO electrode is approximately 58 Ω (Table 3). Anchoring on the surface of a self-assembled layer with an amine group (FTO/APTMS (16 h)) reduces the Rct value by 10 Ω (Figure 2A). As expected, increasing the APTMS deposition time from 16 to 72 h still lowers the surface resistance to 30 Ω. As a result, after 72 h of APTMS deposition, the Rct value was almost doubled (from 58 Ω to 30 Ω) compared to the Rct value for the unmodified FTO electrode. Moreover, we observe the changes in the shape of the Nyquist plot. The diameter of the semicircle decreased, and the linear section lengthened in the low-frequency range. This means that the electron transfer process is faster than the diffusion of the redox system, close to the electrode surface. This effect appears to be due to the increased amount of amine groups on the electrode material, as well as the greater order of the layer. Moreover, as the deposition time lengthened, the capacity of the double layer increased, while the value of the n parameter decreased, which proves the heterogeneity of the FTO/APTMS surface.

In turn, the electrodes covered with the OTMS layer show completely different properties (Figure 2B). For the FTO/OTMS modification (16 h), we observed a curve in the Nyquist plot consisting of a large semicircle and a small linear range at low frequencies. This spectral image illustrates a process limited by the rate of electron transfer through the double layer. As a result, diffusion is more efficient than electron transport. This is because the OTMS layer blocks the access of the [Fe(CN)_6_]^3−/4−^ system from the solution to the electrode surface. However, the shape of the spectrum suggests the presence of defects in the layer that is formed, which is reflected in an increase in the Rct value. The blocking degree is higher with the FTO/OTMS electrode (72 h). Due to the extension of the deposition time, for this modification, the impedance spectrum consists only of a semicircle in the tested frequency range. This demonstrates an even slower charge transfer reaction, due to the formation of a long aliphatic chain layer. The presence of very large semicircles confirms the excellent electrochemical blocking ability of the OTMS layers. The Rct values for the FTO/OTMS electrodes are much higher than for the FTO/APTMS electrodes. After the first modification (FTO/OTMS (16 h)), the electron transfer resistance increased from 58 Ω to approximately 426 Ω. Conversely, after 72 h of deposition (FTO/OTMS (72 h)), the resistance increased up to 1.5 kΩ. A higher Rct value implies a lower double-layer capacity, as evidenced by the data in Table 3.

It is worth emphasizing that the impedance results are compatible with the measurements obtained for cyclic voltammetry. In CV measurements, similarly to EIS measurements, different electrochemical properties of the modified FTO/APTMS and FTO/OTMS electrodes are observed (Figure 2C,D). In the case of the FTO/APTMS modification, the cathode peak shifts slightly towards the positive values of the potential, thus increasing the reversibility of the redox process of the [Fe(CN)_6_]^3−/4−^ system. In total, the ΔE value decreased by 40 mV (Table 3), which confirms that the presence of amino groups on the FTO electrode surface improves the electron transfer process.

Conversely, the deposition of the OTMS layer shows a blocking effect for the same process. After 16 h, the ΔE value increased from 259 to 593 mV, indicating the reduced reversibility of the test system. At the same time, a significant decrease in the intensity of the oxidation current and reduction peaks was noted, by 284 µA and 206 µA, respectively, compared to the unmodified FTO electrode. The further process of surface silanization with OTMS increases the peak separation up to 832 mV, with an approximately 50% reduction in the current response intensity (Figure 2D). This clearly shows that this redox reaction at the FTO/OTMS electrode (72 h) was blocked due to the formation of highly ordered and well-packed self-assembled OTMS layers with low defect density. The moderate blocking effect for the shorter deposition time can be attributed to the formation of a layer with more holes and defects, allowing electron tunneling. The recorded changes confirm that the extent of blocking is different for each modification, which depends primarily on the nature and the method of layer formation, and, thus, on the chemical structure formed on the electrode surface.

### 3.2. Double-Alkoksysilane Modification

The differences observed for the APTMS- and OTMS-modified electrodes prompted us to synthesize the systems resulting from the action of a silane solution with two alkoxysilanes at the same time. The analysis of the impedance spectra shows that the deposition time is the decisive factor in the final surface properties of the modified FTO/APTMS_OTMS electrode. The Nyquist plot shows two different curves, depending on the duration of the FTO electrode silanization process (Figure 3A). The tendency of the changes is completely different compared to the previously described FTO/APTMS and FTO/OTMS modifications. After 16 h of deposition, the shape of the impedance spectrum corresponds to the characteristics of the FTO/APTMS electrodes, while after 72 h, it is similar to the FTO/OTMS electrodes. These changes are reflected in the Rct values. Initially, the resistance slightly decreases to 55 Ω and then increases sharply after the next modification, up to 22.5 kΩ (Table 3). In the case of the modification of FTO/APTMS_OTMS (72 h), due to the shape of the Nyquist plot, a different model of the equivalent circuit, R(QR), was used, disregarding the Warburg impedance representing the diffusion element. In the case of the FTO/APTMS_OTMS (16 h) electrode, the Q value is close to the electrical capacity of the output electrode. The parameters Rct and Q suggest that the cross-linking of the material takes place in a shorter time than in the longer layer-formation process. Thus, extending the modification time may lead to secondary changes on the electrode surface. Again, the results obtained from the EIS measurements correlate with the CV results (Figure 3B). The difference in the potential of the oxidation and reduction peaks of the system [Fe(CN)_6_]^3−/4−^ for the FTO/APTMS_OTMS (72 h) modification is 1.19 V, where in the case of a shorter deposition time, the value of ΔE oscillates around 0.26 V. Together with an increase in the separation of peaks, there is also a dramatic decrease in the intensities of the current response. The intensity of the current for the anode and cathode response for the FTO/APTMS_OTMS electrode (72 h) is three times lower than for the FTO/APTMS_OTMS (16 h) or FTO (Table 3). Hence, the longer deposition of the APTMS and OTMS mixture causes the reorganization of the ordered structure, leading to the formation of a steric hindrance for the access of [Fe(CN)_6_]^3−/4−^ ions to the electrode surface.

The specific electrochemical behavior of the FTO/APTMS_OTMS electrodes prompted us to change the conditions of the experiment. The mixed silane layer has re-formed, but this is now as a result of the two-step surface modification of the FTO electrode (Figure 4).

### 3.3. Two-Step Mixed Alkoksysilane Modification

APTMS and OTMS monosubstrate-modifying solutions were used for the silanization process. We obtained two new electrodes—FTO/APTMS/OTMS and FTO/OTMS/APTMS—using a different order of alkoxysilane deposition. Based on our expectations, the materials modified in this way should show similar conductive properties to the FTO/APTMS_OTMS electrode. It would seem that, regardless of the order of deposited alkoxysilanes, the obtained layers will have an almost identical surface structure and, consequently, the resistance to charge transfer at the electrode/solution interface will be at the same level. The obtained results undoubtedly contradict the expected results. Regardless of the duration of the silanization process, the electrochemical properties of the resulting layers depend essentially on the second deposition step. Despite the fact that a mixed layer of silanes is formed on the surface of the FTO electrode, we observed the dominance of one of the silanes in the impact on the new structural features. In this respect, this effect is comparable to the electrodes obtained as a result of the action of the alkoxysilane mixture (FTO/APTMS_OTMS), where the deposition time was decisive for the predominant proportion of one of the components. On the other hand, for the electrodes obtained at certain stages (FTO/APTMS/OTMS and FTO/OTMS/APTMS), this fact was determined by the substrate participating in the second stage of silanization, as can be concluded based on the electrochemical parameters of the monosubstrate electrodes (FTO/APTMS and FTO/OTMS). Depending on which of the silanes was applied last (APTMS or OTMS), the values of Rct, Q, and ΔE are, respectively, lower or higher than for the one-component modifications (FTO/APTMS or FTO/OTMS) (Table 3). Thus, in the case of the FTO/APTMS/OTMS (16 h) modification, a greater share of the silane layer is attributed to long octyl chains (Figure 4A). The Rct value for this surface is 336.2 Ω, which is only 90.2 Ω lower than the monosubstrate derivative, FTO/OTMS (16 h). This change is influenced by the presence of aminopropyl groups on the surface, which lowers the charge transfer resistances while increasing the capacity of the double layer. APTMS seems to be responsible for organizing the newly formed structure on the surface of the FTO electrode, marking the places where OTMS can be deposited, which determines the lipophilic nature of the surface. For the second modification of FTO/OTMS/APTMS (16 h), the effect of double silanization is different (Figure 4B). Despite the two-stage process, the conductive properties of this electrode are comparable to the FTO/OTMS electrode (16 h). The resistance value is about 440 Ω, which confirms the similarity to the monosubstrate derivative (FTO/OTMS (16 h)). However, the Q value indicates that APTMS contributes, to some degree, to forming this structure, although this is not evident in the Rct value. It is probable that a well-organized OTMS layer blocks access to the free places on the electrode surface. The APTMS molecules are not able to overcome this steric hindrance within 16 h and settle on the surface, between the OTMS chains. Nevertheless, the analysis of impedance spectra confirms this thesis because a longer silanization time significantly reduces the surface resistance by about 340 Ω, which proves the reorganization of the structure. Depositing the APTMS particles on the electrode surface increases their share in the newly formed structure, thus lowering the charge transfer resistance. The effect of the denser packing of silanes on the FTO/OTMS/APTMS electrode surface (72 h) confirms the lower capacity of the double layer, compared to the Q value, after 16 h of deposition. In contrast to the FTO/APTMS/OTMS electrode (72 h), we observed an increase in the charge transfer resistance for the model system [Fe(CN)_6_]^3−/4−^ from 336 to 1664 Ω and a decrease in the double layer capacitance, which confirms the increasing share of OTMS in blocking the surface. Undoubtedly, the impedance results correlate with the CV measurements (Figure 4C,D). The ΔE value represents the above-described electrochemical properties of the obtained electrodes. For the FTO/APTMS/OTMS electrode, the ΔE value almost doubled with increasing deposition time, while for the FTO/OTMS/APTMS modification, we observed a twofold decrease in the ΔE value (Table 3).

Apart from the deposition time of alkoxysilanes, the temperature is undoubtedly another factor influencing the effectiveness of the silanization process. It has been proven that by increasing the reaction temperature, we can simultaneously reduce the deposition time. APTMS and OTMS monosubstrate silanizing solutions were again selected for modification, under changed process conditions. By increasing the temperature to 50 °C and reducing the deposition time to 30 min, two new electrodes were obtained, namely, FTO/APTMS (0.5 h) and FTO/OTMS (0.5 h). The EIS and CV measurements recorded for the new layers were analogous to those obtained after the longer deposition times of APTMS and OTMS (72 h) at room temperature. This proves that the electrochemical properties of the newly formed structures are comparable with those of the FTO/APTMS (72 h) and FTO/OTMS (72 h) electrodes. However, the silanization process in this case is more difficult to control. By simultaneously selecting two variable deposition parameters (time and temperature), from which the resulting structure is conditioned, we do not have the guarantee of obtaining the same defined electrode surface each time. Therefore, we decided to keep the modification at room temperature by changing only the deposition time of the silanes.

The tests showed that the silanized electrode materials were characterized by different stability levels over time. The electrochemical parameters that we took into account were the surface resistance and the reversibility of the redox process. The example of the FTO/APTMS electrode (Figure 5A,B) shows that 6 days after modification, the surface resistance increases significantly, as well as the separation between the oxidation and reduction peaks. In the case of electrodes where OTMS dominates in the structure (Figure 5C,D), this effect could already be noticed on the second day. Due to the fact that the observed changes were significant, we did not set any threshold values, but all tests for the newly formed layers, as well as the subsequent stages of synthesis on the surface, were performed immediately after the deposition process, meaning that the measurement conditions were identical for each electrode.

### 3.4. Wettability

For further characterization, we investigated the wettability of the electrodes modified with alkoxysilanes used for electrochemical tests. The FTO electrodes that were modified with APTMS, PTMS (aliphatic APTMS), and OTMS were measured by means of the contact angle measurements, as shown in Figure 6.

The contact angle (WCA) value for the unmodified FTO electrode is approximately 56°. The hydroxyl groups formed as a result of activation (oxidation) of the substrate are responsible for the hydrophilic nature of the surface. Based on the FTO/APTMS, FTO/PTMS, and FTO/OTMS modifications, we observed that the contact angle changed depending on the structure of the formed silane layer. For the FTO/APTMS electrode, the angle value decreased to 53°, but for the FTO/PTMS and FTO/OTMS electrodes, the value increased significantly to 71° and 90°, respectively. The FTO/OTMS surface shows a higher WCA value than the FTO/PTMS, due to the presence of long aliphatic chains in the structure. On the surface, highly ordered self-assembling OTMS layers with low defect density are formed, giving the electrode a hydrophobic character. Undoubtedly, the FTO/PTMS electrode also has more hydrophobic properties than the FTO/APTMS. In turn, the presence of an amino group in the side chain of the alkoxysilane (FTO/APTMS) completely changes the surface properties of the electrode. Compared to its alkyl derivative (FTO/PTMS), it has a lower angle value, which proves the good wettability of the FTO/APTMS electrode surface. The obtained values of the contact angles are close to the values reported in the literature, even in relation to the alkoxysilane layers formed on other electrode substrates [24,48,49]. However, it should be remembered that the wettability of the surface strongly depends on the nature of the formed silane layer, and, thus, on the conditions in which it was formed, which may cause discrepancies in the given angle values [50].

## 4. Conclusions

Based on the EIS and CV measurements, we assessed the operating time of the modified FTO electrodes, from which one can conclude that the electrochemical parameters of the new materials change over time. The FTO/APTMS and FTO/OTMS/APTMS electrodes show the highest reproducibility of results. Regardless of the deposition time, these electrodes are electrochemically stable for about two days, which suggests that the dominant share of APTMS in the newly formed layers extends the life of the electrode. In the case of other electrodes where the structure is dominated by OTMS, it is necessary to immediately perform planned measurements or the subsequent modification stages immediately after silanization, because their electrochemical parameters change over time. In turn, the stability of the FTO/APTMS_OTMS electrode depends upon the timeframe of silane deposition. A greater dispersion of the measurement results was noted for a longer modification time. In contrast, the FTO/APTMS_OTMS electrode surface (16 h) is reactive for two days. Slight differences in the stability of the resulting structures suggest that the silanized FTO electrodes should not be stored. All tests for the newly formed layers, as well as the subsequent stages of synthesis on the surface, should be performed immediately after the deposition process [51].

The discussion of the results found in the course of the conducted research allows for the conclusion that deposition time plays a key role in the formation of self-organizing silane layers. A longer silanization process leads to a reorganization of the structure. Consequently, the newly formed structure causes the layer to have new properties that differ from the intermediate and initial properties. Apart from the interaction time of the silanizing solution with the surface of the FTO electrode, it is also important whether the solution is monocomponent or binary. In the case of two-step modification with alkoxysilanes, the electrochemical properties of the modified surfaces depend on the second deposition step. Comparing the values of Rct and Q, we can state that the best predisposition to create an ordered SAM structure was shown by the APTMS one-component solution. In turn, the FTO/APTMS_OTMS electrode (72 h) (Figure 7) showed the greatest blocking effect for the electrochemical process of the redox system [Fe(CN)_6_]^3−/4−^. Taking into account the shorter time of the silanization process, the charge transfer resistance at the electrode/solution interface increased in the order:

However, after 72 h of deposition, the direction of the changes was exactly the opposite. Moreover, the modification of the FTO electrode surface, using selected self-assembled layers of organosilane molecules, enables the fine-tuning of their hydrophobic/hydrophilic properties by controlling the side chain length and the nature of the molecule.

This research confirms the high efficiency of the silane groups’ deposition on FTO glass substrates. By selecting the appropriate alkoxysilanes, one can control the surface properties of the electrode materials and, thus, regulate the electron transfer process at the electrode/solution interface.

## Figures and Tables

**Figure 1 materials-16-00309-f001:**
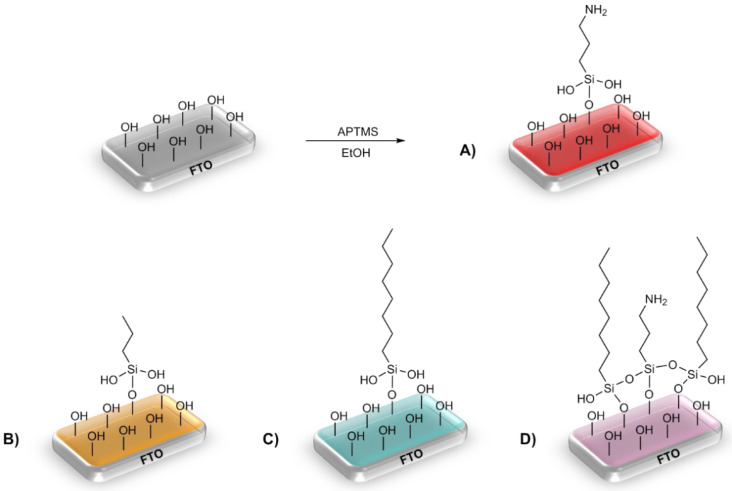
Schematic diagram of the silanization process, with an example of the surface structure of the modified electrodes—FTO/APTMS (**A**), FTO/PTMS (**B**), FTO/OTMS (**C**), and FTO/ATMS_OTMS (**D**).

**Figure 2 materials-16-00309-f002:**
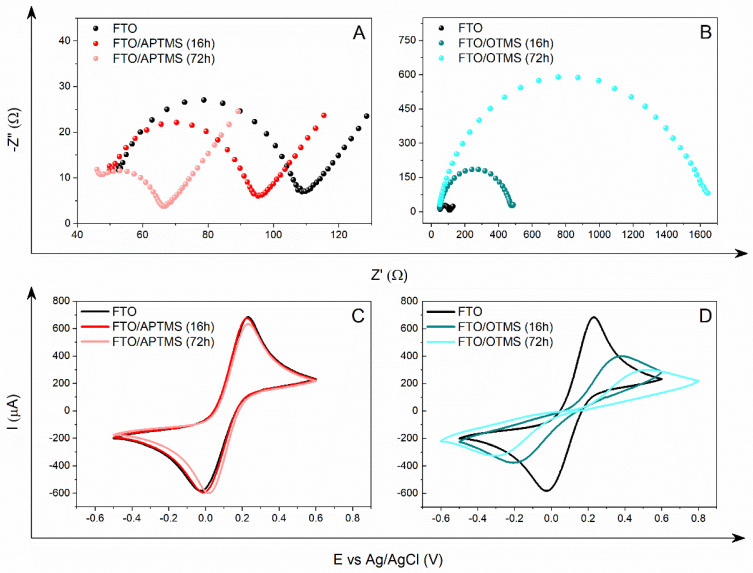
Nyquist plots (**A**,**B**) and voltammograms (**C**,**D**) showing the modification of the FTO electrode with APTMS (**A**,**C**) and OTMS (**B**,**D**) after different silanization times (16 h, 72 h), recorded in 0.5 M KCl (EIS) and Na_2_SO_4_ (CV) containing a 5 mM redox system [Fe(CN)_6_]^3−/4−^.

**Figure 3 materials-16-00309-f003:**
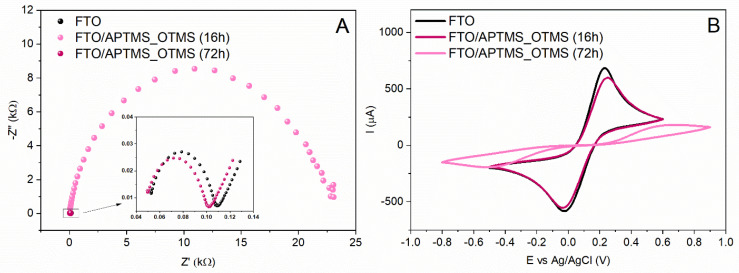
Graph of the Nyquist (**A**) and cyclic voltammetry (**B**) values obtained for the modified FTO/APTMS_OTMS electrodes after different deposition times (16 h, 72 h), recorded in a 0.5 M aqueous solution of KCl (EIS) and Na_2_SO_4_ (CV), containing a 5 mM redox system [Fe(CN)_6_]^3−/4−^.

**Figure 4 materials-16-00309-f004:**
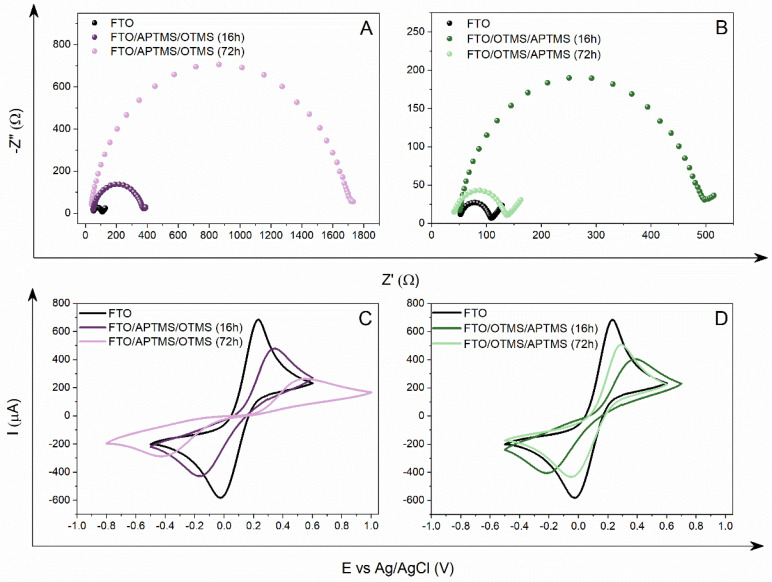
Graphs of the Nyquist (**A**,**B**) and cyclic voltammetry (**C**,**D**) values obtained for modified electrodes as a result of two-stage FTO/APTMS/OTMS (**A**,**C**) and FTO/OTMS/APTMS (**B**,**D**) silanization, after different time depositions (16 h, 72 h), recorded in a 0.5 M aqueous solution of KCl (EIS) and Na_2_SO_4_ (CV), containing a 5 mM redox system [Fe(CN)_6_]^3−/4−^.

**Figure 5 materials-16-00309-f005:**
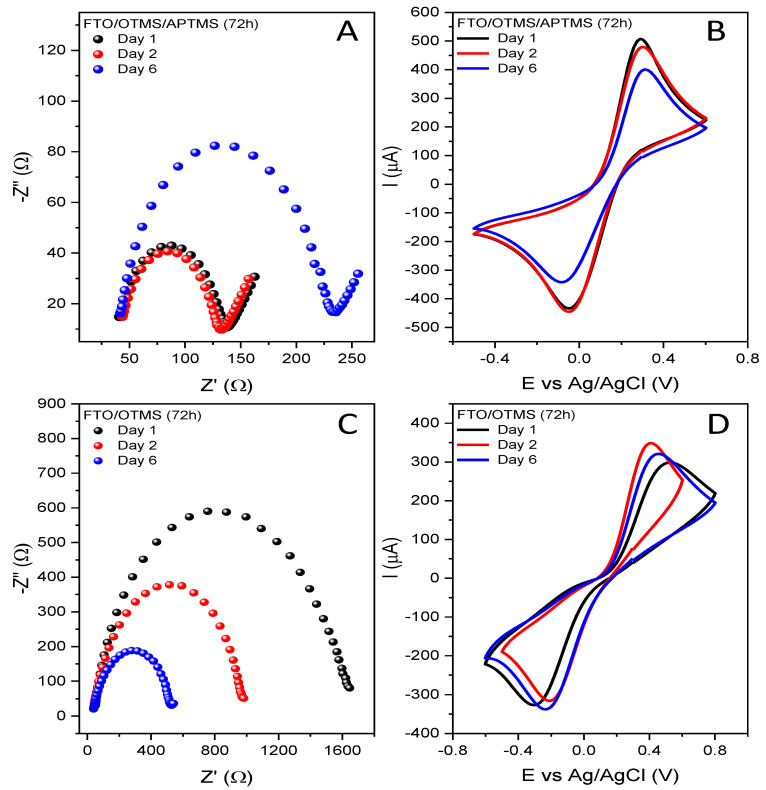
Graphs of Nyquist (**A**,**C**) and cyclic voltammetry (**B**,**D**) obtained for modified electrodes as a result of two-stage FTO/OTMS/APTMS (**A**,**B**) silanization and one-stage FTO/OTMS (**C**,**D**) after different time of storage (1–6 days), recorded in a 0.5 M aqueous solution of KCl (EIS) and Na_2_SO_4_ (CV) containing a 5 mM redox system [Fe(CN)_6_]^3−/4−^.

**Figure 6 materials-16-00309-f006:**
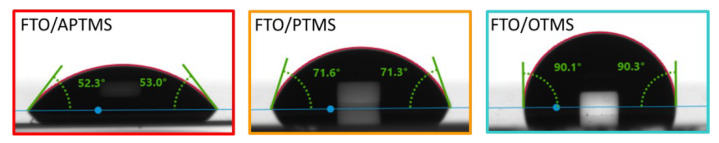
Pictures of the water contact angle (WCA) measurements of modified FTO/APTMS, FTO/PTMS, and FTO/OTMS electrodes.

**Figure 7 materials-16-00309-f007:**
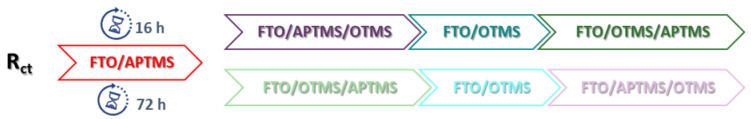
Schematic presentation of changes in charge transfer resistance for the investigated electrode materials.

**Table 1 materials-16-00309-t001:** Silanes used in the surface modification process.

Silane	Acronym	Molecular Structure
3-aminopropyltrimethoxysilane	APTMS	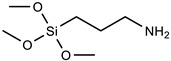
Trimethoxy(propyl)silane	PTMS	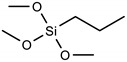
trimethoxy(octyl)silane	OTMS	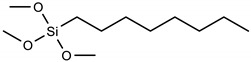

**Table 2 materials-16-00309-t002:** Conditions for the silanization process on the surface of the oxidized FTO electrode.

Silane	Reaction Conditions (Temperature, Time)	Electrode	Second Reaction Conditions (Temperature, Time)	Electrode
APTMS	23 °C; 16 h	FTO/APTMS (16 h)	23 °C; 16 h	FTO/APTMS/OTMS (16 h)
	23 °C; 72 h	FTO/APTMS (72 h)	23 °C; 72 h	FTO/APTMS/OTMS (72 h)
	50 °C; 0.5 h	FTO/APTMS (0.5 h)		
PTMS	50 °C; 0.5 h	FTO/PTMS (0.5 h)		
OTMS	23 °C; 16 h	FTO/OTMS (16 h)	23 °C; 16 h	FTO/OTMS/APTMS (16 h)
	23 °C; 72 h	FTO/OTMS (72 h)	23 °C; 72 h	FTO/OTMS/APTMS (72 h)
	50 °C; 0.5 h	FTO/OTMS (0.5 h)		
APTMS and OTMS	23 °C; 16 h	FTO/APTMS_OTMS (16 h)		
	23 °C; 72 h	FTO/APTMS_OTMS (72 h)		

**Table 3 materials-16-00309-t003:** Electrochemical parameters of FTO electrodes after modification with alkoxysilanes, obtained on the basis of CV and EIS measurements by fitting the experimental data with the equivalent R(Q(RW)) or R(QR) circuits.

Electrode	R_ct_ [Ω]	Q [µF]	n	W [µSs^½^]	Chi^2^	E_a_ [V]	E_k_ [V]	ΔE [V]	I_a_ [µA]	I_k_ [µA]
FTO	57.87	6.65	0.91	0.012	3.27 × 10^−4^	0.233	−0.026	0.259	683.3	−583.2
FTO/APTMS (16 h) *	48.46	8.28	0.88	0.012	6.19 × 10^−4^	0.226	−0.012	0.238	676.3	−594.2
FTO/APTMS (72 h) *	30.74	11.23	0.79	0.012	2.69 × 10^−4^	0.233	0.016	0.217	632.3	−600.9
FTO/OTMS (16 h) *	426.40	5.69	0.89	0.015	7.31 × 10^−4^	0.384	−0.209	0.593	399.2	−376.9
FTO/OTMS (72 h) *	1545.00	3.52	0.84	0.004	3.47 × 10^−4^	0.521	−0.311	0.832	298.5	−326.8
FTO/APTMS_ OTMS (16 h) *	55.05	8.03	0.88	0.012	5.85 × 10^−4^	0.244	−0.016	0.260	609.1	−591.4
FTO/APTMS_ OTMS (72 h) **	22,450.00	0.25	0.86	-	8.04 × 10^−4^	0.676	−0.514	1.190	180.3	−195.1
FTO/APTMS/ OTMS (16 h) **	336.20	6.35	0.87	-	1.05 × 10^−3^	0.346	−0.167	0.513	478.2	−429.3
FTO/APTMS/ OTMS (72 h) *	1664.00	1.36	0.89	0.007	2.27 × 10^−4^	0.546	−0.423	0.969	262.6	−288.1
FTO/OTMS/ APTMS (16 h) *	439.30	4.39	0.89	0.009	4.91 × 10^−4^	0.388	−0.219	0.607	403.4	−406.6
FTO/OTMS/ APTMS (72 h) *	98.19	6.86	0.88	0.010	8.54 × 10^−4^	0.293	−0.047	0.340	506.6	−433.5

* R(Q(RW)); ** R(QR); ΔE = E_a_ − E_k_.

## Data Availability

The data presented in this study are available on request from the corresponding author.

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
