# Peer review of "Controlled Silanization of Transparent Conductive Oxides as a Precursor of Molecular Recognition Systems"

_materials, 2022, doi:10.3390/ma16010309_

Round 1

Reviewer 1 Report

This study reported changes in electrical properties and surface energy characteristics depending on the surface treatment results of FTO. In addition, the effect of surface treatment time and order was observed. However, the SAM layers deposited on the FTO surface are not new and the results are predictable from the existing papers. Also, the analysis is simplistic and the facts derived from the experimental results are not interesting.

Author Response

List of changes

Ms. Ref. No.:  materials-2050136

Title: Controlled Silanization of Transparent Conductive Oxides as a Precursor of Molecular Recognition Systems

Authors: Anna Domaros *, Dorota Zarzeczańska, Tadeusz Ossowski, Anna Wcisło *

Materials (ISSN 1996-1944)

First of all, we would like to sincerely thank the Reviewer for the effort put in the review of our article, and for all the insightful comments and remarks that have allowed us to improve it. A revision of the manuscript has been performed. Below, I present a detailed list of changes introduced to the text in response to particular points raised by the Reviewer.

REVIEWER #1:

This study reported changes in electrical properties and surface energy characteristics depending on the surface treatment results of FTO. In addition, the effect of surface treatment time and order was observed. However, the SAM layers deposited on the FTO surface are not new and the results are predictable from the existing papers. Also, the analysis is simplistic and the facts derived from the experimental results are not interesting.

Answer: Thank you for taking the time and effort to review our work. We appreciate any comments to improve the quality of our manuscript. We are aware that many papers describing self-organizing layers on electrode surfaces, including FTO electrodes, have already been published. Nevertheless, our work focuses on silanization conditions in terms of their properties. So far, no work has been published presenting variable conditions, such as temperature and time, and more importantly, taking into account two-component modifying mixtures (alkoxysilanes). Also, two-step silanization processes have not been described in the literature so far. Therefore, it seems to us that such a work will find its audience among the readers of the Materials.

Action: No action.

I hope that the presented explanations and introduced corrections will be satisfactory to the Reviewer. Thank you again for your time and effort to improve our manuscript.

Please do not hesitate to contact me in case of any doubts or questions.

Yours sincerely

Anna Wcisło

Reviewer 2 Report

I would suggest to specify in the abstract: - which alcoxysilanes were studied;-  which conditions of modifications were varied.

From the introduction the readers don't have the information concerning other attempts of silanization of the materials under study. It's necessary to clarify in the introduction  whether the authors are the only researchers, who made the experiments with FTO electrodes. Definitely, the studies of relative materials are known. However the authors briefly refer to a few of them in the very last section of the discussion.

The abbreviation FTO should be explained earlier, in the line 37, where we meet it for the first time.

Author Response

List of changes

Ms. Ref. No.:  materials-2050136

Title: Controlled Silanization of Transparent Conductive Oxides as a Precursor of Molecular Recognition Systems

Authors: Anna Domaros *, Dorota Zarzeczańska, Tadeusz Ossowski, Anna Wcisło *

Materials (ISSN 1996-1944)

First of all, we would like to sincerely thank the Reviewer for the effort put in the review of our article, and for all the insightful comments and remarks that have allowed us to improve it. A revision of the manuscript has been performed. Below, I present a detailed list of changes introduced to the text in response to particular points raised by the Reviewer.

REVIEWER #2:

I would suggest to specify in the abstract: - which alcoxysilanes were studied;-  which conditions of modifications were varied.

Answer: Thank you very much for this suggestion. Appropriate changes have been made in the manuscript.

Action: Page 1 line 11: “The aim of the work was to modify the surface of electrodes based on transparent conductive oxides with the use of selected alkoxysilanes (3-aminopropyltrimethoxysilane, trimethoxypropylsilane and trimethoxyoctylsilane).”

Page 1 line 15: “Here we prove that not only the structure of alkoxysilanes was taken into account, but also the conditions of the modification process – reaction conditions (time and temperature), double alkoxysilanes modification, mono- and binary component modification mixture.”

From the introduction the readers don't have the information concerning other attempts of silanization of the materials under study. It's necessary to clarify in the introduction whether the authors are the only researchers, who made the experiments with FTO electrodes. Definitely, the studies of relative materials are known. However the authors briefly refer to a few of them in the very last section of the discussion.

Answer: Naturally, the process of silanization of the surface of electrode materials is a known modification technique. Our literature research has shown that various conditions for this process have been used in the previously published works. They mainly differed in time, temperature and composition of the modifying reagent. However, there was no work that would explain how these specific conditions affect the properties of the obtained layers, and thus what parameters to choose to achieve the desired effect. The Reviewer rightly pointed out that we did not emphasize this enough in the Introduction. Thanks to the Reviewer's comment, we have added an appropriate fragment of the text and additional citations.

Action: Page 2 Line 59: The covalent attachment of specific biomolecules to the surface of a conducting mate-rial is of great importance for the development of sensors based on the molecular recogni-tion. The F functionalization of thea surface by inclusion in the structure of a compound with specific properties requires the presence of an intermediate organic layer. In the case of oxide surfaces, such as an FTO electrode, organosilanes are the most common com-pounds mediatingwhich mediate in the anchoring of the molecules to the substrate [9,34–38]. The silanization process is the first step towards the construction of a biosensor based on the FTO electrode. Reports available in the literature show a wide variety of conditions used in this stage, starting from the choice of the solvent itself, silane concentration, to the temperature and time of the electrode incubation in silane solutions [23,24,30,39]. Howev-er, there is no explanation how these specific conditions affect the properties of the ob-tained layers, and thus what parameters to choose to achieve the desired effect. Therefore, we decided  to optimize the deposition conditions of the selected group of alkoxysilanes and their mixtures were optimized with . Pparticular attention was paid to the settling time and the reaction temperature at which the reaction was carried out. Fo-cusing on the selection of appropriate conditions, our the aim was to obtain the best pos-sible repeatability of the silane deposition process on for the surface of the FTO electrode surface. In addition, we verified the stability of the modified electrodes over time was checked in order to determine their working time and, consequently, their susceptibility to further modification stages.

The abbreviation FTO should be explained earlier, in the line 37, where we meet it for the first time.

Answer: It was our mistake. We didn't notice it. Thank you very much for your remark.

Action: Page 1 line 34: “Fluorine doped tin oxide (FTO) electrodes are considered a very promising material due to their greater stability in atmospheric conditions and higher temperature resistance than indium tin oxide (ITO) electrodes.”  

I hope that the presented explanations and introduced corrections will be satisfactory to the Reviewer. Thank you again for your time and effort to improve our manuscript.

Please do not hesitate to contact me in case of any doubts or questions.

Yours sincerely

Anna Wcisło

Reviewer 3 Report

In this manuscript, the authors reported EIS, CV, and wettability results of the FTO substrates functionalized through three alkoxysilanes. The contents of the experimental and analysis results are rich and comprehensive, and the discussion is quite complete. In general, the manuscript is well-written, and the results could be attractive in the related community. However, the following points should be addressed before accepted for publication in the Materials: 

1.     There is no testing result about molecular recognition. The title should not mention any about that, and should be modified according to the contents of the manuscript.

2.     The reason of adopting transparent electrodes is not clear. In Lines 30-32 the authors said “However, the combination of two different properties characteristic of transparent conductive oxides, i.e. optical transparency and high electrical conductivity, makes this material extremely interesting for further research [14,15].”. However, ref [14] is a general review on flexible transparent conductive electrode, while ref [15] is about flexible transparent conductor for strain sensor application. The authors have to make the reason clearer in the Introduction.

3.     Abbreviation of FTO should be defined in Line 37, rather than Line 46. Same as EIS in Lines 100 and 105.

4.    Lines 123-139 should be in the Introduction. And Lines 141-171 and Lines 175-176 should be in the Materials and Methods.

5.    The Conclusion has to be rewrite and rephrase because there are no stability test results in this manuscript. In Lines 396-400 the authors claimed “In the case of other electrodes, where the structure is dominated by OTMS, it is necessary to immediately perform planned measurements or subsequent modification stages immediately after silanization, because their electrochemical parameters change over time. In turn, the stability of the FTO/APTMS_OTMS electrode depends on the time of silane deposition.” And in Lines 137-139: “In addition, the stability of the modified electrodes over time was checked in order to determine their working time and, consequently, their susceptibility to further modification stages.”

6.     Continued, regarding the stability issue, the authors have to clarify how long they took prior to performing tests (after samples as made). In this way comparable results in the Results and Discussion are validated.

Author Response

List of changes

Ms. Ref. No.:  materials-2050136

Title: Controlled Silanization of Transparent Conductive Oxides as a Precursor of Molecular Recognition Systems

Authors: Anna Domaros *, Dorota Zarzeczańska, Tadeusz Ossowski, Anna Wcisło *

Materials (ISSN 1996-1944)

First of all, we would like to sincerely thank the Reviewer for the effort put in the review of our article, and for all the insightful comments and remarks that have allowed us to improve it. A revision of the manuscript has been performed. Below, I present a detailed list of changes introduced to the text in response to particular points raised by the Reviewer.

REVIEWER #3:

In this manuscript, the authors reported EIS, CV, and wettability results of the FTO substrates functionalized through three alkoxysilanes. The contents of the experimental and analysis results are rich and comprehensive, and the discussion is quite complete. In general, the manuscript is well-written, and the results could be attractive in the related community. However, the following points should be addressed before accepted for publication in the Materials:

1.There is no testing result about molecular recognition. The title should not mention any about that, and should be modified according to the contents of the manuscript.

Answer: We fully understand the concern and doubts of the Reviewer. Nevertheless, we would prefer not to change the title of the manuscript. Our goal was to indicate that this type of systems most often serve as precursors in the creation of molecular recognition systems. These are elements that connect the electrode material with the sensory element. We believe that the title formulated in this way will attract the attention of a wider group of readers – especially those dealing with the creation of sensors based on conductive oxides. We believe that it will be interesting for them, as it will allow for more effective control of the properties of the final product.

Action: No Action.

2.The reason of adopting transparent electrodes is not clear. In Lines 30-32 the authors said “However, the combination of two different properties characteristic of transparent conductive oxides, i.e. optical transparency and high electrical conductivity, makes this material extremely interesting for further research [14,15].”. However, ref [14] is a general review on flexible transparent conductive electrode, while ref [15] is about flexible transparent conductor for strain sensor application. The authors have to make the reason clearer in the Introduction.

Answer: In fact, as indicated by the Reviewer, we described these elements in the work quite briefly. This may mislead the reader and cause misunderstanding. We have added more references and additional explanation in the text of the manuscript.

Action: Page 1 Line 34 “Fluorine doped tin oxide (FTO) electrodes are considered a very promising material due to their greater stability in atmospheric conditions and higher temperature resistance than indium tin oxide (ITO) electrodes. Moreover, this material is chemically inert, mechanically resistant, and has a high resistance to physical abrasion [16,17]. Both FTO and ITO electrodes find practical applications in a wide range of devices, among others, to provide transparent conductive coatings in touch panels, flat screens, aircraft cockpit windows, or plasma monitors. Thin oxide films are also used in the production of organic light-emitting diodes (OLEDs) and in solar cells [18,19]. In addition, due to their unique features, these materials provide a good basis for the modification of surface properties [16,20–24]. Moreover, they are also used in classical electroanalytical measurements, as working electrodes in the determination of a wide range of electroactive compounds [25].”

  1. Abbreviation of FTO should be defined in Line 37, rather than Line 46. Same as EIS in Lines 100 and 105.

Answer: It was our mistake – we didn't notice it. Thank you very much for your remark.

Action: Page 1 line 34: “Fluorine doped tin oxide (FTO) electrodes are considered a very promising material due to their greater stability in atmospheric conditions and higher temperature resistance than indium tin oxide (ITO) electrodes.”  

Page 5 line 153: “Using the electrochemical impedance spectroscopy (EIS) technique enabled the determi-nation of the charge transfer resistance (Rct), which is a measure of the resistance of self-organizing silane layers to electron transfer from the solution to the electrode surface.” 4.    Lines 123-139 should be in the Introduction. And Lines 141-171 and Lines 175-176 should be in the Materials and Methods.

Answer: Thank you very much for this remark. Appropriate changes have been made and marked in the text.

Action: Changes were made to the manuscript as suggested by the Reviewer.

  1. The Conclusion has to be rewrite and rephrase because there are no stability test results in this manuscript. In Lines 396-400 the authors claimed “In the case of other electrodes, where the structure is dominated by OTMS, it is necessary to immediately perform planned measurements or subsequent modification stages immediately after silanization, because their electrochemical parameters change over time. In turn, the stability of the FTO/APTMS_OTMS electrode depends on the time of silane deposition.” And in Lines 137-139: “In addition, the stability of the modified electrodes over time was checked in order to determine their working time and, consequently, their susceptibility to further modification stages.” Continued, regarding the stability issue, the authors have to clarify how long they took prior to performing tests (after samples as made). In this way comparable results in the Results and Discussion are validated.

Answer: The reviewer is right, this information should be included in the manuscript. We have included the relevant information in the content of the article. We also thank you for your thorough assessment.

Action: Page 13, line 448: The tests showed that the silanized electrode materials were characterized by differ-ent stability over time. The electrochemical parameters that we took into account were the surface resistance and the reversibility of the redox process. The example of the FTO/APTMS electrode (Fig. 5 A,B) shows that after 6 days from the modification the sur-face resistance increases significantly, as well as the separation between the oxidation and reduction peaks. In the case of electrodes where OTMS dominates in the structure (Fig. 5 C,D), this effect could be noticed already on the second day. Due to the fact that the observed changes were significant, we did not set any threshold values but all tests for the newly formed layers, as well as the subsequent stages of synthesis on the surface, were performed immediately after the deposition process, so that the measurement conditions were identical for each electrode.

Figure 5. Graphs of Nyquist (A, C) and cyclic voltammetry (B, D) obtained for modified electrodes as a result of two-stage FTO/OTMS/APTMS (A, B) silanization and one-stage FTO/OTMS (C, D) after different time of storage (1-6 days) recorded in a 0.5 M aqueous solution of KCl (EIS) and Na2SO4 (CV) containing a 5 mM redox system [Fe(CN)6]3-/4-.”

I hope that the presented explanations and introduced corrections will be satisfactory to the Reviewer. Thank you again for your time and effort to improve our manuscript.

Please do not hesitate to contact me in case of any doubts or questions.

Yours sincerely

Anna Wcisło

Round 2

Reviewer 1 Report

I agree that the search for new molecular recognition systems is important. I hope that the research in this paper can be expressed in detail how it can contribute to molecular recognition.

Reviewer 3 Report

Thanks for the revisions.